# Cu_2_ZnSnS_4_ (CZTS) for Photoelectrochemical CO_2_ Reduction: Efficiency, Selectivity, and Stability

**DOI:** 10.3390/nano13202762

**Published:** 2023-10-15

**Authors:** Yijia Zhang, Shujie Zhou, Kaiwen Sun

**Affiliations:** 1School of Photovoltaic and Renewable Energy Engineering, University of New South Wales (UNSW), Sydney, NSW 2052, Australia; yijiazhang@zju.edu.cn; 2School of Chemical Engineering, University of New South Wales (UNSW), Sydney, NSW 2052, Australia

**Keywords:** CZTS, CO_2_ reduction, photoelectrochemical

## Abstract

Massive emissions of carbon dioxide (CO_2_) have caused environmental issues like global warming, which needs to be addressed. Researchers have developed numerous methods to reduce CO_2_ emissions. Among these, photoelectrochemical (PEC) CO_2_ reduction is a promising method for mitigating CO_2_ emissions. Recently, Cu_2_ZnSnS_4_ (CZTS) has been recognized as good photocathode candidate in PEC systems for CO_2_ reduction due to its earth abundance and non-toxicity, as well as its favourable optical/electrical properties. The performance of PEC CO_2_ reduction can be evaluated based on its efficiency, selectivity, and stability, which are significantly influenced by the photocathode materials. As a result, researchers have applied various strategies to improve the performance of CZTS photocathodes, including band structure engineering and surface catalytic site engineering. This review provides an overview of advanced methods to enhance the PEC systems for CO_2_ reduction, focusing on CZTS.

## 1. Introduction

The rising demand for fossil fuels and energy consumption has become a critical concern, as it carries the potential to trigger an energy crisis [1] and result in an alarming rise in atmospheric carbon dioxide (CO_2_) levels, further exacerbating global warming [2]. To tackle these pressing challenges, researchers have identified the reduction of CO_2_ to chemical fuels as a promising and feasible solution [3]. This intricate chemical process converts CO_2_ into carbon-containing chemicals such as carbon monoxide (CO), methanol (CH_3_OH), formate (HCOOH), and methane (CH_4_) [3], garnering attention from industries and the scientific community for decades [4].

Recently, researchers have explored diverse technologies for CO_2_ reduction, with photocatalytic (PC) reduction, electrocatalytic (EC) reduction, and photoelectrochemical (PEC) reduction [5] emerging as the primary means to convert CO_2_ into valuable chemicals. Among these methods, PC reactions driven by solar energy have gained traction as an eco-friendly and efficient approach, characterized by simplicity and energy efficiency (Figure 1a [6]). In this process, the photocatalytic material absorbs sunlight to generate electron–hole carriers, used for CO_2_ reduction [7]. It is worth mentioning that the photocatalytic reduction reaction can be performed in either a gas atmosphere or a liquid. There have been many papers recently showing good performance in photocatalytic reduction in gas atmospheres [8,9,10,11]. On the other hand, EC reduction directly utilizes electrons from external electricity to drive the reaction (Figure 1b [6]). Especially in recent developments, electrocatalytic reduction is connected with a photovoltaic (PV) component, namely, PV + EC, representing a combined strategy that uses photovoltaic systems to absorb solar energy and generate electricity, which is then used to drive electrochemical cells to drive CO_2_ reduction. With the urgent need to address rising CO_2_ levels in the atmosphere and the potential to boost sustainable fuel production, PV + EC is now one of most promising strategies [8]. Despite this advantage, EC CO_2_ reduction still requires an external bias to drive the reaction. In contrast, PEC systems combine the merits of both PC and EC reactions (Figure 1c [6]). Photogenerated electrons and holes on the photoelectrode are directed under the influence of a moderate external bias, or even without external bias, to the catalyst surface to drive the CO_2_ reduction, thereby allowing the PEC system to convert solar energy into chemical energy. Although the current efficiency of the PEC system may not rival that of EC reduction, it boasts the advantage of not requiring an intricate infrastructure. In comparison to PC reduction, PEC reduction exhibits superior efficiency and facilitates straightforward product separation [12]. Notably, the PEC system stands out due to its ability to facilitate product separation and manipulate the reaction mechanism and kinetics for product selectivity tuning with the assistance of an externally applied voltage [13] (see Table 1).

One promising candidate for an eco-friendly photocathode material in CO_2_ reduction is the p-type semiconductor Cu_2_ZnSnS_4_ (CZTS), which possesses a suitable bandgap of approximately 1.5 eV [14]. CZTS is an I-II-IV-VI4 quaternary compound semiconductor with advantages such as an excellent light absorption ability, low cost, abundance on earth, and environmental friendliness [15]. CZTS usually exists as kesterite-type structure, which is more thermodynamically stable. The crystal structure diagram of kesterite CZTS is shown in Figure 2; the different colours represent different elements, and the bandgap depends on the synthesis method and specific composition. There are a lot of different fabrication methods of CZTS, like the electrochemical deposition method, the electron beam evaporation method, the magnetron sputtering method, the spray pyrolysis method, the sol–gel method, and thermal evaporation deposition [16]. Researchers from the University of New South Wales recently demonstrated CZTS solar cells with the highest efficiency of 11.4% [17,18]. Moreover, compared with other semiconductors, CZTS exhibits superior capability for PEC CO_2_ reduction due to its more favourable conduction band position (Figure 3). These characteristics underscore the potential of CZTS films for PEC CO_2_ reduction [19]. Consequently, extensive research efforts are dedicated to enhancing the performance of CZTS thin films in PEC CO_2_ reduction.

## 2. Basic Principles of PEC CO_2_ Reduction

### 2.1. Thermodynamics and Kinetics of CO_2_ Reduction

CO_2_ adopts a linear molecule structure, composed of a lone carbon © atom positioned between two oxygen (O) atoms. The bonds connecting carbon and oxygen (C=O) within the CO_2_ molecule are established through a covalent bonding mechanism, characterized by a mutual sharing of electrons between these constituent atoms. This shared electron arrangement results in the formation of robust bonds with a strong bonding energy of 750 kJ mol^−1^, contributing to the molecular structure’s inherent resistance to degradation [22]. Figure 2 illustrates a comparison of band values among the semiconductors used in the process of PEC CO_2_ reduction, alongside potential products. Typically, CO_2_ reduction products include carbon monoxide (CO), formic acid (HCOOH), methanol (CH_3_OH), ethanol (C_2_H_5_OH), and hydrocarbons. The activation of CO_2_ (CO_2_ to *CO_2_^−^) requires a high potential (−1.9 V), as shown in Figure 3, which is much higher than that of water splitting. Therefore, a high applied potential is commonly used for PEC CO_2_ reduction. This difference primarily stems from variations in the stability and electronic configurations of the reactants and products. Additionally, the cleavage energy that is required to break the C=O bond is notably high, demanding greater activation energy than water splitting. Consequently, CO_2_ reduction necessitates more complex conditions [23].

From a kinetic perspective, the synthesis of carbon-based compounds typically requires the transfer of a larger number of electrons compared to the hydrogen evolution reaction (HER), which involves the transfer of only two electrons [5]. The process of converting CO_2_ into other compounds involves diverse reaction pathways, consisting of sequences of chemical reactions from the initial material (CO_2_) to end products. These pathways involve different quantities of electrons (2e^−^, 4e^−^, 6e^−^, 8e^−^, etc.), intensifying the intricacy of reaction kinetics (rate of reaction). Due to this complexity, the meticulous control of the reaction kinetics to enhance the efficiency and selectivity of CO_2_ reduction stands as a substantial challenge within this field [24].

### 2.2. Configurations of PEC System

There are four types of PEC technology systems based on different photoactive semiconductors: the photocathode-driven system (Figure 4a), the photoanode-driven system (Figure 4b), the photoanode and photocathode system (Figure 4c), and the solar cell tandem system (Figure 4d) [21]. Consequently, the selection of appropriate photoactive semiconductors as photocathodes or photoanodes is crucial for the effectiveness of the PEC system [25]. Typically, a photocathode-driven PEC system for CO_2_ reduction is a three-electrode configuration, as shown in Figure 5, involving three main components: the semiconductor photocathode itself (potentially with a co-catalyst), a counter electrode for water oxidation, and a reference electrode. Together, these components create an environment where the light-absorbing semiconductor photocathode can drive the reduction of CO_2_ while maintaining a balanced reaction through the water oxidation at the counter electrode. Therefore, it is essential for a photocathode semiconductor to possess a suitable bandgap to be activated under light exposure [26]. Additionally, the photogenerated electrons and holes must have the minimum thermodynamic energies necessary for the intended application, so as to enable surface redox reactions [13]. Regulating the properties of semiconductor-based photocathodes, including the bandgap, band alignment, and co-catalyst, is crucial for improving their overall performance. This can be achieved through various strategies, such as adjusting the bandgap to increase light absorption, modifying band structure to facilitate redox reactions, and improving photocharge transport to enhance the surface charge transfer [13].

### 2.3. General Parameters to Evaluate the PEC CO_2_ Reduction

The performance of PEC CO_2_ reduction can be evaluated using several key parameters. Typical metrics in PEC CO_2_ reduction systems are as follows:Faradaic efficiency (FEs) [22]: This metric quantifies the ratio of the desired PEC system’s actual yield to the theoretical yield, measuring how efficiently electrical energy is converted into a chemical reaction during CO_2_ reduction. It is typically expressed as a percentage that is calculated using the following formula:

(1)$$FE\left(\%\right)=\frac{\alpha n_{\left(A\right)}F}{Q}$$
where α is the number of electrons needed to reduce CO_2_ to yield one molecule of *A*. $n_{\left(A\right)}$ is the molar quantity of *A*, F is the Faraday constant, and Q denotes the total charge through the circuit in the process.

Partial photocurrent density ($j_{A})$ [27]: $j_{A}$ is used to evaluate the catalytic activity of the photocathode toward one product (*A*), which is identified by multiplying the photocurrent density $j_{ph}$ with the corresponding FE(A) (Equation (1)). In PEC CO_2_ reduction, a higher photocurrent density indicates a higher rate of CO_2_ reduction reaction, which is desirable for achieving higher efficiency.



(2)
$$j_{A}=j_{ph}\times FE\left(A\right)\;$$



These efficiency metrics can be determined experimentally by analysing the current–voltage characteristics, examining the type of products, and comparing them with the total current passed during the CO_2_ reduction process. It is worth noting that the efficiency of CZTS in CO_2_ reduction may also be affected by factors such as catalyst influence, electrode design, and operating conditions [28,29]. Therefore, it is crucial to consider these factors alongside the inherent characteristics of CZTS when evaluating efficiency.

## 3. Applications of CZTS in PEC CO_2_ Reduction

Despite the advantages of CZTS [29], there are challenges associated with the implementation of CZTS in PEC CO_2_ reduction, including its slow charge transport, low catalytic activity, and instability [30]. Arial et al. demonstrated the potential of CZTS in conjunction with a metal-complex electrocatalyst for selective CO_2_ photoreduction in 2011 [31]. Subsequently, researchers have been actively optimizing various methods to improve the performance (efficiency, selectivity, and stability) of CZTS-based PEC systems for CO_2_ reduction [32]. The strategies for improving the performance include band structure engineering to enhance charge separation and light absorption, and catalytic site regulation to enhance the catalytic activity and selectivity of active sites. Band structure engineering encompasses activities like modifying the band gap, adjusting energy levels, and creating p-n junctions. On the other hand, catalytic site regulation involves processes such as altering surfaces, introducing functional groups to surfaces, utilizing co-catalysts, and employing electrocatalysis.

### 3.1. Efficiency

The key prerequisite for successfully developing CZTS-based PEC systems for CO_2_ reduction is the high faradaic efficiency. The pioneering work in 2011 demonstrated the ability of CZTS photocathodes, modified with a ruthenium complex polymer (MCE), to reduce CO_2_ to formate [31].

In a pivotal study, Yoshida et al. explored the capabilities of a pristine CZTS film for use as a photocathode in PEC CO_2_ reduction, employing both theoretical calculations and experimental evaluations [33]. They put the CZTS film under visible light irradiation with an applied potential of −0.8 V (vs. Ag/AgCl), as depicted in Figure 6. Notably, CZTS exhibited a higher cathodic photocurrent under CO_2_ bubbling than under N_2_, suggesting its potential for CO_2_ reduction under visible light. For the theoretical aspect, they applied first-principles calculations to determine the position of the valence band maximum (VBM) and conduction band maximum (CBM), as well as to estimate the bandgap. The band edge positions with respect to the vacuum level were also determined using a slab model of CZTS, which implies that CZTS can drive CO_2_ reduction to generate various organic fuel molecules. In combination with theoretic calculations and experimental investigations, their study demonstrated the capability of pure CZTS films for CO_2_ reduction as photocathodes through the PEC system and showed their excellent stability, even after 48 h.

Ohno et al. further improved the efficiency of PEC CO_2_ reduction by introducing a p-n heterojunction formed between bare CZTS and n-type (CdS and In_2_S_3_) buffer layers, which is a good example of band structure engineering strategy. They prepared CZTS films using the sol–gel spin-coating method [34], followed by the deposition of two different n-type buffer layers (CdS and In_2_S_3_) on the CZTS film using the chemical bath deposition (CBD) method. The CO_2_ reduction experiments were conducted in a gastight three-electrode configuration cell with a double compartment, using a 0.1 M NaHCO_3_ electrolyte solution. The results revealed that CdS/CZTS achieved the highest Faradaic efficiency at 83.5% in total, surpassing the efficiencies of bare CZTS and In_2_S_3_/CZTS.

Zhou et al. improved PEC CO_2_ reduction efficiency through the heat treatment (HT) of a CZTS/CdS heterojunction [35,36]. They prepared pristine CZTS thin films using the co-sputtering and sulfurization processes on Mo-coated soda–lime glass (SLG) substrates, and the CdS was then coated on the CZTS thin films via the CBD method [19]. In their comparative analysis, the researchers investigated the effect of HT on photocathode charge carrier transfer and PEC CO_2_ reduction performance [29], where pristine CZTS and CZTS/CdS without HT were compared with CZTS/CdS subjected to HT in an air atmosphere using a hot plate at 270 °C for 10 min, as shown in Figure 7. They found that the heat treatment process facilitated elemental diffusion at the CZTS/CdS interface, optimizing band alignment. Consequently, this led to an enhanced charge transfer, and facilitated the electron transfer and improved the PEC CO_2_ reduction performance through higher band bending and reduced energy loss.

Although HT can minimize carrier loss in CZTS thin films, it also poses the risk of CZTS decomposition and element loss [36], emphasizing the importance of selecting appropriate HT conditions to obtain high-performance films. In 2022, Heidari et al. proposed a method that involved high-temperature heat treatments in a sulphur vapor or hydrogen sulphide gas atmosphere for a shorter duration, which is considered effective in preventing the elements of the thin film from escaping [37]. In their study, CZTS nanocrystals were prepared using the solvothermal technique and subsequently immersed in an ethanol solution, followed by the creation of CZTS nanocrystal films using the simple dip coating technique [38,39]. To investigate the influence of different HT conditions (temperature and duration) on the PEC water splitting performance of the films, the CZTS thin films were examined in a three-electrode setup under an N_2_ environment with a flow rate of 40 mL min^−1^. Their study demonstrated that low-temperature heat treatments could not fully decompose the surfactant polyvinylpyrrolidone (PVP) in the films, while temperatures exceeding 400 °C led to the occurrence of SnS_x_ impurities, resulting in a reduced PEC performance, as shown in Figure 8. Interestingly, they found that fast heat treatments for CZTS thin films yielded better PEC performance compared to slow heating. Although their study primarily focused on water splitting, the potential applicability of their method to CO_2_ reduction was also highlighted.

A PEC system requires a lower external bias voltage for driving the reaction, necessitating the use of a photocathode with a more negative conduction band minimum (CBM) [40]. To this end, some researchers have explored the potential of high-bandgap Cu_2_ZnGeS_4_ (CZGS) semiconductors, which share the same kesterite structure as CZTS, for PEC CO_2_ reduction. In 2019, Ikeda et al. fabricated both CZTS and CZGS thin film samples using spray deposition to compare their PEC performance in CO_2_ reduction [17]. The bandgaps of CZGS and CZTS were determined through polyethersulfone (PES) measurement, with values of 2.16 eV and 1.48 eV, respectively. The relatively high CBM potential of CZGS is crucial for facilitating CO_2_ reduction by inducing sufficient overpotential for the reaction. Despite all modified films showing appreciable photocurrents, CZGS displayed lower photocurrents compared with the CZTS-based electrode. This study highlights the need for a photocathode with a more negative CBM to drive the CO_2_ reduction reaction in the PEC system.

In conclusion, CZTS demonstrates promise as a photocathode for CO_2_ reduction, and researchers have employed different strategies to improve its efficiency. These strategies include modifying the band alignment, enhancing the charge transfer, and optimizing reaction conditions [41]. However, addressing challenges related to selectivity and stability remains a priority in advancing CZTS-based CO_2_ reduction technologies toward practical implementation. Further research is required to overcome these hurdles and unlock the full potential of CZTS in CO_2_ reduction.

### 3.2. Selectivity

The selectivity of PEC CO_2_ reduction refers to its capacity to selectively convert carbon dioxide (CO_2_) into desired chemical products while minimizing the generation of unwanted byproducts, which is crucial for determining the overall efficiency of the CO_2_ reduction process [42]. In essence, it quantifies the efficiency of converting CO_2_ into a specific target product, such as carbon monoxide (CO), formic acid (HCOOH), methanol (CH_3_OH), ethanol (C_2_H_5_OH), and hydrocarbons [43]. Table 2 lists the standard reduction potentials for different CO_2_ reduction pathways and their corresponding products. Notably, the required reduction potentials of various products are closely situated, presenting thermodynamic challenges in achieving selective product formation [44].

Thus, it is crucial to consider the pathway of PEC CO_2_ reduction for improving selectivity, since different pathways contribute to different products. The main chemical products of CO_2_ reduction are C_1_ and C_2_. The C_1_ pathway is simple, in which CO as the intermediate is transformed into methane through the formation of *CHO or *COH species. However, C_2_ and C_2_^+^ products are much more complex due to the need for suitable catalysts. Normally, the catalytic site regulation strategy has been used to improve the selectivity of PEC CO_2_ reduction.

In 2011, Arial et al. first investigated the semiconductor/metal-complex electrocatalyst (SC/MCE) hybrid semiconductor as a photocathode for selective CO_2_ reduction to formate (HCOO^−^) [30]. The improved current efficiency of formate formation (EEF) over the modified CZTS photocathode underscores the effectiveness of catalytic site regulation. Their research centres on the use of CZTS and CZTSSe as potential photocathodes. These materials are particularly enticing due to their composition of earth-abundant elements, offering a sustainable and cost-effective alternative to other semiconductors like InP. However, the true highlight of the study is the modification of these photocathodes with a ruthenium complex polymer, known as a metal-complex electrocatalyst (MCE). This modification significantly enhances the selectivity of the CO₂ reduction reaction.

Furthermore, by using the surface modification of catalytic site regulation, Kong et al. investigated the special bonding properties of CO_2_ under both exposed metal and sulphur terminations of CZTS. They explored the microscopic mechanism underlying the formation of CO, HOOH, and CH_4_ [45]. They proposed four pathways (Path I–IV) for HCOOH and CO formation, considering different intermediates and steps. In the case of metal termination, HCOOH is formed through Path II, while CO is primarily generated via Path III. The production of HCOOH and CO typically involves a two-electron (2-ele) process and is usually generated through four pathways:

Path I: CO_2_^−^ → COOH^−^ →HCOOH^−^ → HCOOH

Path II: CO_2_^−^ → HCOO^−^ → HCOOH

Path III: CO_2_^−^ → COOH^−^ → CO^−^ + OH^−^ → CO^−^ → CO (g)

Path IV: CO_2_^−^ → CO^−^ + O^−^ → CO^−^ → CO (g)

The onset potential for HCOOH production is 0.23 V, while CO formation via COOH^−^ requires a higher onset potential of 1.66 eV. The desorption energies indicate the easy removal of the intermediates from the metal termination. Moreover, on the sulphur termination, HCOOH and CO preferentially follow Path I and Path III, respectively, with onset potentials of 0.34 V and 0.37 V. The desorption energies also suggest the efficient removal of these products from the sulphur termination. The first intermediate (COOH^−^, HCOO^−^, or CO^−^ + O^−^) plays a crucial role in determining the reaction pathways, with different mechanisms linked to CO_2_ adsorption states. Therefore, the results demonstrate the high selectivity of HCOOH towards the metal termination, while an increase in the exposed sulphur termination leads to the appearance of CH_4_ and CO products. This study provides new insights into the selectivity of PEC CO_2_ reduction with CZTS photocathodes.

In addition, in a comparative experiment conducted by Zhou et al., efforts were made to modulate the selectivity of CO_2_ reduction products upon HT under different atmospheres, particularly, CO, CH_3_CH_2_OH, and CH_3_OH [19]. Their results showed that CZTS/CdS with HT in nitrogen (HN) contributed to higher selectivity toward CO, and CZTS/CdS with HT in air (HA) produced higher amounts of alcohols, as shown in Figure 9.

The abovementioned studies extensively explored the selectivity for PEC CO_2_ reduction using CZTS, aiming to convert CO_2_ into desired chemical products while minimizing unwanted byproducts. Key factors influencing selectivity include the band gap value and the CO_2_ reduction pathway. Hence, researchers identified surface vacancy engineering and different high-temperature treatments as effective strategies to optimize PEC CO_2_ reduction selectivity.

### 3.3. Stability

The stability of the CZTS photocathode refers to its ability to maintain its catalytic activity and performance over extended periods of operation. Ensuring that the CZTS layer in the photocathode remains stable is essential for supporting efficient light absorption and charge separation [46]. At a molecular level, CZTS demonstrates stability due to its crystal structure and the arrangement of its constituent elements. However, like many semiconductor materials, CZTS materials can experience degradation under prolonged exposure to electrolytes, leading to changes in their optical and electronic properties [47]. Therefore, research that is focused on enhancing the stability of CZTS under illumination holds significant importance.

Notably, the study by Yoshida et al. demonstrated the remarkable stability of the CZTS photocathode for PEC CO_2_ reduction reactions during long-term visible light irradiation [33], with consistent photocurrent performance and product generation over 48 h, as shown in Figure 10. The CZTS thin film was prepared using a spin-coating technique to ensure a highly pure CZTS phase. Their results showed that the pristine PEC CO_2_ reduction reaction using a CZTS thin film as the photocathode was stable and generated both CO and H_2_.

## 4. Conclusions and Outlook

In conclusion, this review provides a comprehensive exploration of CZTS-based PEC CO_2_ reduction, delving into its efficiency, selectivity, and stability. The investigation casts a spotlight on key aspects such as the heterojunction structure, preparation methodologies, and operating conditions, highlighting their pivotal roles in shaping the performance landscape. Notably, research on CZTS-based PEC CO_2_ reduction remains nascent, with the exact reduction pathway and underlying mechanisms yet to be fully unveiled. Concurrently, unresolved challenges demand attention, including issues like sluggish charge transport, inadequate catalytic activity, and inherent instability. Despite notable strides in recent years, it remains imperative to underscore specific future perspectives in this burgeoning research arena, given the increasing significance of CZTS-based PEC CO_2_ reduction technology in the realm of solar-to-fuel conversion.

Firstly, there exists a compelling need to significantly enhance the efficiency of CZTS-based PEC CO_2_ reduction. This goal can be achieved through the meticulous optimization of energy band alignment, thereby boosting the participation of electrons in the CO_2_ reduction process and surmounting potential energy barriers. These efforts are crucial for advancing the performance of CZTS-based PEC CO_2_ reduction, thus unfurling the full potential of sustainable energy conversion. For instance, the incorporation of more compatible heterojunctions and favourable photosensitisers proves to be instrumental in enhancing light absorption for CZTS-based PEC CO_2_ reduction. Moreover, achieving precise interface regulation, finely tuning charge transfer kinetics, and designing spatially separated directional pathways for electron–hole transportation are pivotal for effectively separating and transporting charge carriers in the PEC CO_2_ reduction reaction. These strategies collectively contribute to optimizing the overall efficiency of CZTS-based PEC CO_2_ reduction, holding immense promise in advancing solar-to-fuel conversion technologies.

Secondly, it is vital to explore novel photoactive semiconductors and elucidate the intricate mechanisms underlying the pathway for CO_2_ reduction, as this holds the key to achieving elevated levels of selectivity. The identification and in-depth investigation of new photoactive semiconductor materials open doors to uncovering distinctive properties and potential routes for CO_2_ reduction. Furthermore, delving deeper into the fundamental mechanisms underlying the reduction process paves the way for targeted modifications and refinements to enhance selectivity. Therefore, exploring co-catalysts and deciphering CO_2_ reduction mechanisms emerge as a vital thrust toward advancing selectivity in CZTS-based PEC CO_2_ reduction.

Thirdly, the spotlight shifts to another pivotal facet—the compelling need to fortify CZTS-based PEC CO_2_ reduction against its inherent low stability. CZTS, as a non-oxide semiconductor, is susceptible to photo corrosion, thereby compromising its stability over extended periods. This challenge erects a formidable hurdle in realizing sustained and efficient CO_2_ reduction. Hence, surmounting the stability challenge becomes a linchpin in reinforcing the long-term effectiveness and resilience of CZTS-based PEC CO_2_ reduction systems. In this trajectory, strategies ranging from surface passivation to the application of protective coatings and the scrupulous management of operational conditions emerge as potent contributors to taming photo-induced degradation. Together, these strategies converge to amplify the overall stability of CZTS photocatalysts, thereby laying the foundation for the practical and long-lasting integration of CZTS into the realm of CO_2_ reduction applications.

CZTS exhibits considerable promise as a photocatalyst for CO_2_ reduction, owing to its favourable energy bandgap, desirable electronic properties, and the abundant availability of its constituent raw materials. Ongoing research initiatives predominantly revolve around tackling the challenges posed by its sluggish charge transport, limited catalytic activity, and inherent instability. By improving the efficiency and selectivity of CZTS-based systems, significant strides can be made toward the realization of pragmatic and sustainable CO_2_ conversion technologies. The continuous exploration and advancement of CZTS thin films, coupled with innovations in catalyst design and system optimization, play a pivotal role in propelling the field of PEC CO_2_ reduction. PEC CO_2_ reduction will be applied in the production of renewable fuels, the production of high-value chemicals, and energy storage in the future. Although some achievements have been made, some challenges such as the selectivity towards target products as well as system configuration or production separation still need to be addressed. These advancements offer potential solutions to the overriding concerns of the energy crisis and global warming, heralding a substantial stride toward a future that is rooted in sustainability.

## Figures and Tables

**Figure 1 nanomaterials-13-02762-f001:**
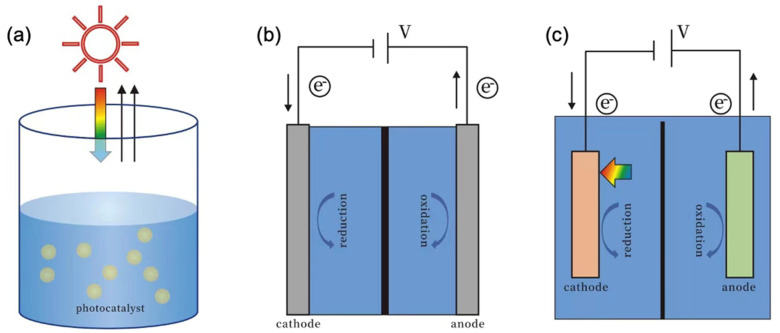
General setup [6] for (**a**) photocatalytic reaction; (**b**) electrocatalytic reaction; (**c**) photoelectrochemical reaction.

**Figure 2 nanomaterials-13-02762-f002:**
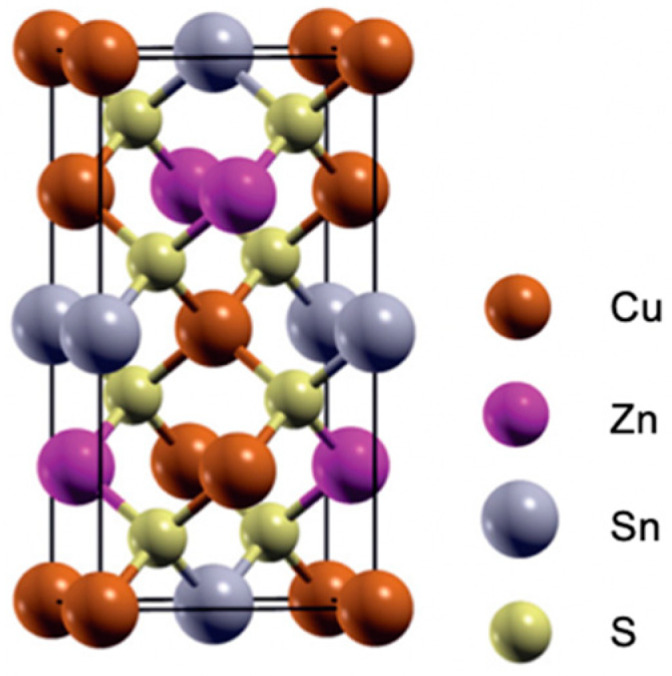
Crystal structure diagram of kesterite. Reprinted with permission from [20]. Copyright 2012 AIP Publishing.

**Figure 3 nanomaterials-13-02762-f003:**
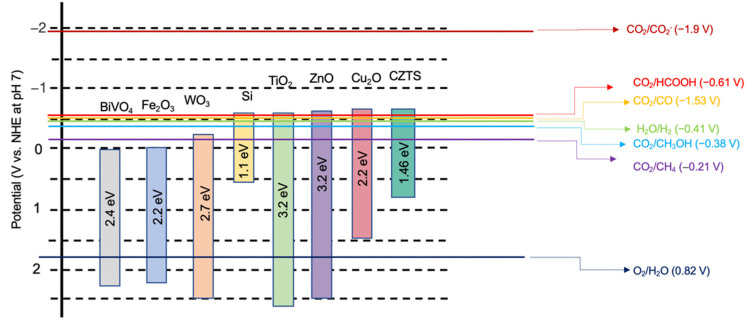
The comparison of band gap values for semiconductors used in PEC CO_2_ reduction. Reprinted with permission from [21]. Copyright 2022 American Chemical Society.

**Figure 4 nanomaterials-13-02762-f004:**
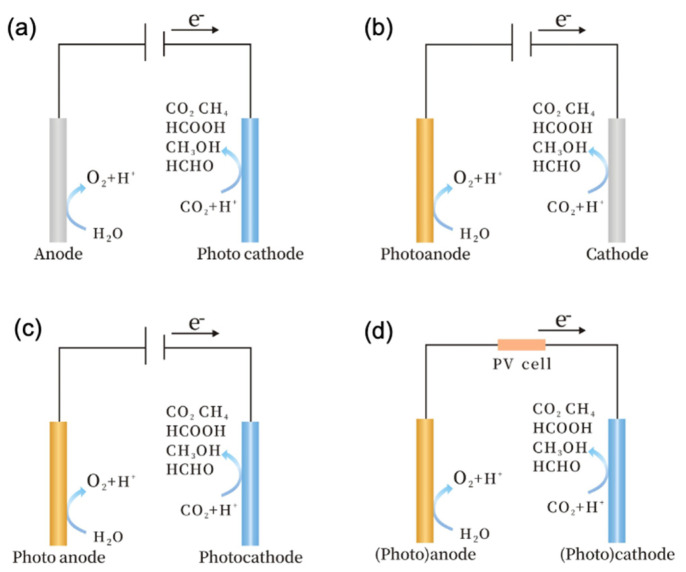
Schematic illustration for the PEC CO_2_ reduction system, including (**a**) photocathode-driven system, (**b**) photoanode-driven system, (**c**) photocathode and photoanode-driven system, and (**d**) hybrid photosystem with the solar cell. Reprinted with permission from [21]. Copyright 2022 American Chemical Society.

**Figure 5 nanomaterials-13-02762-f005:**
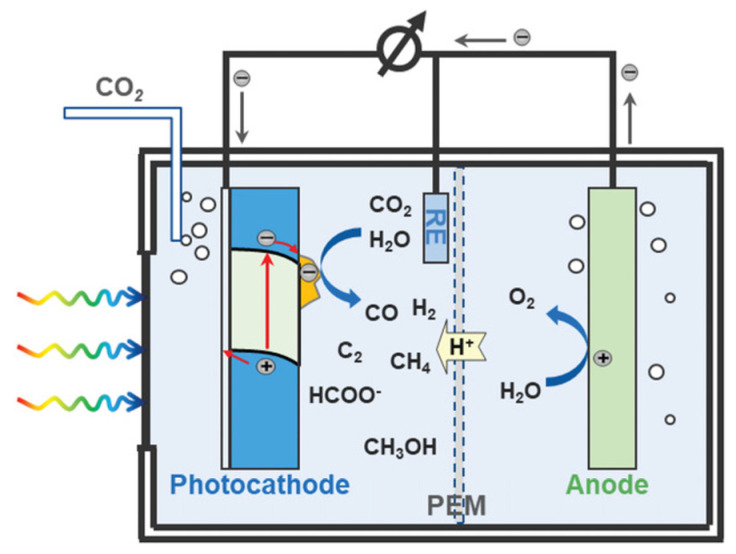
Schematic of the PEC CO_2_ reduction reaction cell with a photocathode in a three-electrode system. Reprinted with permission from [22]. Copyright 2022 John Wiley and Sons.

**Figure 6 nanomaterials-13-02762-f006:**
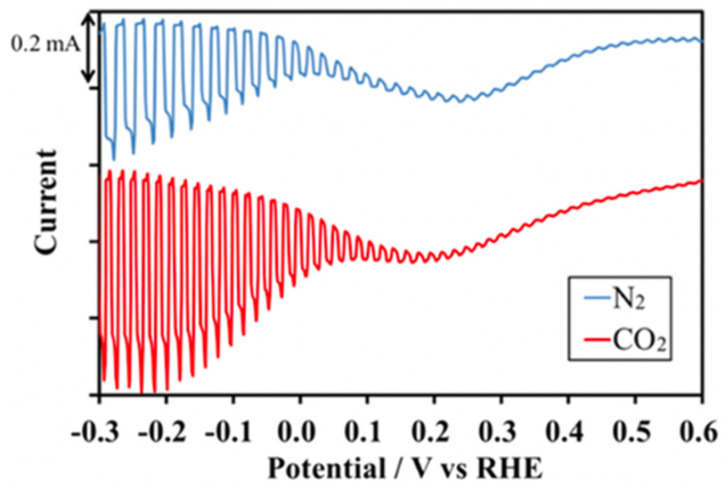
Photocurrent of the CZTS electrode in 0.1 M NaHCO_3_ aq under chopped visible light irradiation (λ > 420 nm) by bubbling N_2_ or CO_2_ gas. The scanning speed of the potential was −10 mV/s, and the electrode size was 1.5 × 1.5 cm. Reprinted with permission from [33]. Copyright 2018 American Chemical Society.

**Figure 7 nanomaterials-13-02762-f007:**
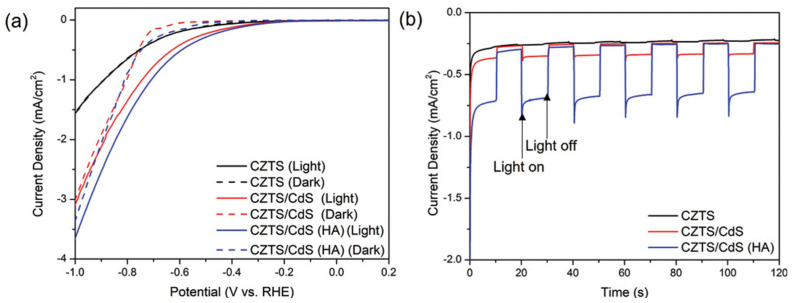
(**a**) LSV curves of CZTS, CZTS/CdS, and CZTS/CdS (HA) with and without illumination. (**b**) Chronoamperometric *I*–*t* curves of CZTS, CZTS/CdS, and CZTS/CdS (HA) under chopped illumination at −0.6 V versus RHE. Reprinted with permission from [19]. Copyright 2021 John Wiley and Sons.

**Figure 8 nanomaterials-13-02762-f008:**
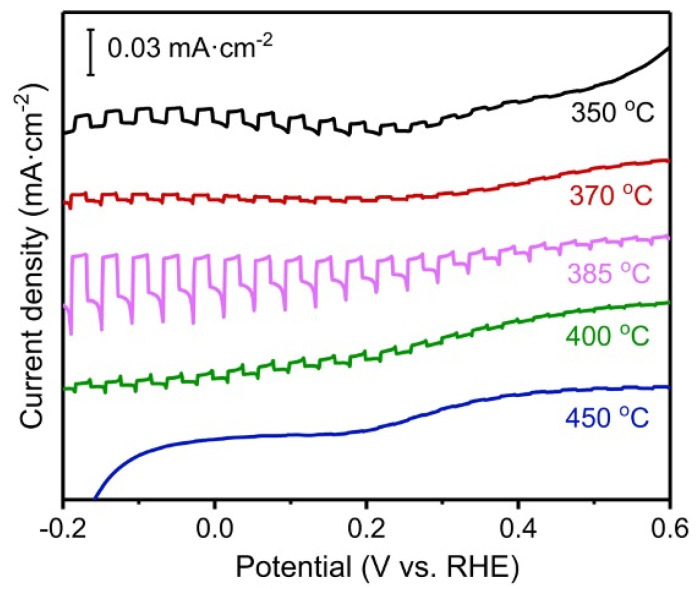
Photocurrent densities of CZTS films after heat treatment at different temperatures. Reprinted with permission from [37]. Copyright 2023 John Wiley and Sons.

**Figure 9 nanomaterials-13-02762-f009:**
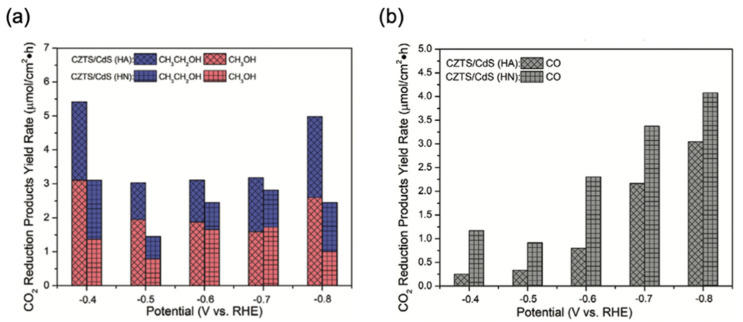
(**a**) The corresponding CO_2_ reduction product (CH_3_CH_2_OH and CH_3_OH) yield rates of CZTS/CdS (HA) and CZTS/CdS (HN). (**b**) The corresponding CO_2_ reduction product (CO) yield rates of CZTS/CdS (HA) and CZTS/CdS (HN) [19]. Copyright 2021 American Chemical Society.

**Figure 10 nanomaterials-13-02762-f010:**
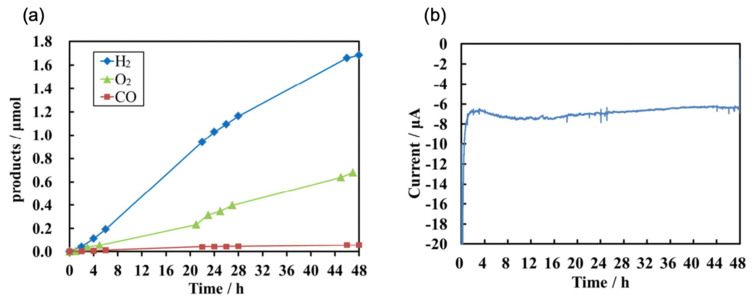
(**a**) The photocatalytic performance of a CZTS electrode under 48 h of irradiation for the generation of H_2_, CO (cathodic side), and O_2_ (anodic side). (**b**) Photocurrent stability of CZTS under 48 h of continuous irradiation. Reprinted with permission from [33]. Copyright 2018 American Chemical Society.

**Table 1 nanomaterials-13-02762-t001:** Similarities and differences between photocatalytic reaction, electrocatalytic reaction, and photoelectrochemical reaction [13].

	Photocatalytic (PC) System	Electrocatalytic (EC)	Photoelectrochemical (PEC)
Incident light	All direction	None	Two directions
Catalyst type	Semiconductor material	Metals, metal oxides, and other conductive materials	Semiconductor material
Energy input	Solar energy	Electricity	Solar energy with/without bias
Product collection	Products separator required	Collected from the separated chamber	Collected from the separated chamber

**Table 2 nanomaterials-13-02762-t002:** Formal redox potential for CO_2_ reduction [21].

Reaction	E0/VSHE (Standard Hydrogen Electrode)	Product
${CO}_{2}+e^{-}\to {CO}_{2}^{-}$	−1.9 V	Charged ions
${CO}_{2}+2H^{+}+2e^{-}\to HCOOH$	−0.61 V	Formic acid
${CO}_{2}+2H^{+}+2e^{-}\to CO+H_{2}O$	−0.53 V	Carbon monoxide
${2H}_{2}O+2e^{-}\to H_{2}+{2OH}^{-}$	−0.41 V	Hydrogen
${{CO}_{2}+5H}_{2}O+6e^{-}\to {CH}_{3}OH+{6OH}^{-}$	−0.38 V	Methanol
${CO}_{2}+8H^{+}+8e^{-}\to {CH}_{4}+{2H}_{2}O$	−0.21 V	Methane

## Data Availability

No new data were created or analyzed in this study. Data sharing is not applicable to this article.

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
