# Peer review of "Cu2ZnSnS4 (CZTS) for Photoelectrochemical CO2 Reduction: Efficiency, Selectivity, and Stability"

_nanomaterials, 2023, doi:10.3390/nano13202762_

Round 1

Reviewer 1 Report

The authors present a review article about the use of (Cu2ZnSnS4) (CZTS) for the photoelectrochemical reduction of CO2.

I think the paper can be published after revisions according to the following points:

1 - Abstract, line 43: "external bias" should be changed into "moderate external bias"

2 - In table 1, row 2, column 4: one direction should be changed into both directions in the case of a a photoactive anode.

3 - I think that at the beginning of the review the authors should at least provide a brief description of the preparations methods for CZTS (hydrothermal, CBD etc..) and a figure with the structure of CZTS. The should also specify that CZTS contains Cu(I) centers and therefore is not too stable. All these aspects should be briefly described before the presentation of the main research results involving CO2 reduction.

For instance the authors describe the nature of the CO2 chemical bond on page 3, lines 70-73. This description is, in my opinion, not necessary since I imagine that people interested in CO2 reduction should at least know the structure and chemical properties of the molecule.

4 - Page 3, lines 78-80: The phrase is not clear and should be re-written. What are the reactions corresponding to the reported values for free energy variations?

5 - Page 5, lines 166-175: The whole paragraph is confusing. It is difficult to understand who performed the theoretical calculations and who the experiments. It should be rephrased.

6 - Page7, line 232: The authors mention the kesterite structure for CZTS, although this is not described or mentioned at the beginning of the review.

7 - Page 8, lines 284-288: The charge of ionic species in the reactions should be written as apex.

8 - Page 10, line 331: The authors write, in the conclusions, that they have described the preparation methods for CZTS. I don't think this is true, and as I wrote before a paragraph should be dedicated to this aspect.

The quality of English should also be improved, but since English is not my native language I cannot be very specific. I think that the English should be improved.

Reviewer 2 Report

This manuscript reviews a photoelectrochemical method, which is one of the common co2 reduction methods, aimed at addressing the global environmental problem of co2 emission reduction. In particular, the use of Cu2ZnSnS4 (CZTS) as a photocathode is discussed. Various aspects and perspectives of CZTS photocathode such as activity, selectivity, stability and so on are reviewed. Recent successes and research results are shown. This review is interesting and written in a concise and understandable language for the reader. However, we would like to suggest the authors to improve the manuscript. Suggestions:

1)Add to the "introduction" a comparison of the efficiency of photocatalytic CO2 reduction in gas atmosphere and liquid. Recently, there are quite a number of papers where photocatalytic CO2 reduction in gas atmosphere show good performance, e.g.:  

https://doi.org/10.1016/j.jhazmat.2020.124019

https://doi.org/10.3390/nano13142030

https://doi.org/10.3390/catal11010047

https://doi.org/10.1016/j.jallcom.2022.164012

2)The discussion and literature references in section "3.3 Stability" need to be expanded. Compared to previous sections, little information is provided.

3) In the section "Conclusions and Outlook" I would like to see in which real sectors of economy it is possible to introduce the considered method of electrochemical reduction of CO2. At the same time what advantages this method will have from the point of view of further perspective.

Finally, Nanomaterials is a suitable journal, provided the paper is accepted for publication.

Reviewer 3 Report

The manuscript of Zhang et al describes the photoelectrochemical application of the semiconductor Cu2ZnSnS4 in the CO2 reduction reaction.

The review is well structured, comprehensive and easy to understand, so it is possible to recommend it for publication.

Minor:

Because the authors made the comparison among three different approaches (PC, EC, and PEC), at least a short discussion of photovoltaic plus electrochemical (PV+EC) must be added, because it is nowadays the most promising strategy (ACS Energy Lett. 2020, 5, 6, 1996–2014).

Reviewer 4 Report

This manuscript summarizes the recent progress regarding Cu2ZnSnS4 (CZTS)-based photoelectrochemical (PEC) system for CO2 reduction.  The reviewer considered that the manuscript could be acceptable after the following minor issues are addressed by the authors.

1.              There have been reported many chemical systems and materials intended for light-driven CO2 reduction, including organic metal complexes, photocatalytic or PEC metal complex-semiconductor hybrid systems, and photovoltaics (PVs) + electrolyzer.  These famous concepts should be briefly overviewed, and the novelty and significance of CZTS-based systems compared to these competitors should be clarified.

2.              In section 3.3, the origin of the stability of CZTS was unclear.

3.              The manuscript contained several typographical errors.  Please carefully check the manuscript before submission.  (e.g., the “2” in “CO2” was sometimes not subscribed; a space was missing from “creatingp-n junction” on line 157.)

Round 2

Reviewer 1 Report

The authors have accepted alla my comments and suggestions. This review paper is improved compared to the previous version and can be published in the present form.

The paper needs a minor English editing.